# Evaluation of Anthracnose Resistance in Pepper (*Capsicum* spp.) Genetic Resources

**Na-Young Ro [1,*], Raveendar Sebastin [1], On-Sook Hur [1], Gyu-Taek Cho [1], Bora Geum [1], Yong-Jik Lee [2] and Byoung-Cheorl Kang [3]**

[1] National Agrobiodiversity Center, National Institute of Agricultural Sciences, Rural Development Administration, Jeonju 54874, Korea; raveendars@gmail.com (R.S.); oshur09@korea.kr (O.-S.H.); gtcho@korea.kr (G.-T.C.); gbora27@korea.kr (B.G.)

[2] HANA SEED, Anseong 456-841, Korea; kamsay45@gmail.com

[3] Department of Plant Science, Plant Genomics and Breeding Institute and Vegetable Breeding Research Center, College of Agriculture and Life Sciences, Seoul National University, Seoul 08826, Korea; bk54@snu.ac.kr

* Correspondence: nonanona@korea.kr

**Abstract:** Anthracnose (*Colletotrichum* spp.), is one of the major yield losing fungal disease in both pre- and post-harvest stage of pepper (*Capsicum* spp.) production worldwide. Among the *Colletotrichum* spp., *C. acutatum* has strong pathogenicity, which infects both immature and mature pepper fruit leads to severe economic losses in pepper production. Inheritance of anthracnose disease resistance was evaluated with 3738 pepper genetic resources which was collected from different countries and conserved at Korean genebank. The resistance analysis against pepper anthracnose (*C. acutatum*) was performed on detached mature green and red fruits under laboratory conditions by spray (non-wounding) and microinjection (wounding) inoculation methods. In the primary screening, about 261 accessions were appeared to be resistant against *C. acutatum* in spray inoculation. The resistant accessions were further evaluated with microinjection (wounding) inoculation method using the fungal (*C. acutatum*) isolate of pepper anthracnose. There were highly significant differences in the disease severity and distribution of disease rating scale, considering all the sources has significant genetic variation. Finally, the anthracnose resistant pepper accessions have been validated with cleaved amplified polymorphic sequence (CAPS) and high-resolution melting (HRM) markers in which, the CAPS and HRM marker analysis showed four types of genotypes such as resistant (R), susceptible (S), heterozygous (H) and Unidentified type (UT) or not detection. The *Capsicum* accessions showing high level of resistance to the pathogen could be used as source material in breeding programs for resistance to anthracnose disease.

**Keywords:** anthracnose resistance; *Capsicum*; *Colletotrichum* spp.; pepper genetic resources

## 1. Introduction

The *Capsicum* L. (Solanaceae) represents a diverse plant group contains a large number of cultivated species as well as wild species that are grown for their fruits, and are an important vegetable consumed throughout the world. Approximately, 25 *Capsicum* species have been cultivated extensively [1] and being used as food flavoring, pharmaceutical ingredient, coloring agent and in many other innovative ways [2]. Systemically, the genus *Capsicum* was classified by its flower, fruit structure and chromosome number [3]. Similarly, *Capsicum* species have been divided into three complexes such as, *C. annuum*, *C. baccatum* and *C. pubescens* complexes based on cytogenetics and cross fertility [4]. However, the wild ancestor of *Capsicum* species remains unclear; due to some wild species have a predominant chromosome numbers [5,6]. Wild species of *Capsicum* are important sources of genetic diversity and reservoirs of genes for breeding programs of cultivated pepper [7]. Hence, conservation of wild genotypes and screening for novel allele is an increasing

priority to make modern cultivars which gradually substitutes the landraces to increase productivity [8].

Anthracnose, is one of the serious fungal disease in pepper fruit caused by *Colletotrichum* spp. such as *C. acutatum*, *C. capsici*, and *C. gloeosporioides*, leads to significant yield losses worldwide [9]. However, in Korea the *C. acutatum* species complex is a most significant causal pathogen of the disease, which infects both immature and mature pepper fruits [10]. Typical anthracnose symptoms on pepper fruit includes sunken necrotic tissues, with concentric rings of acervuli which reduce fruit quality [11]. In general, the anthracnose disease was controlled by using chemical fungicides which might have negative impact on human health and pollute the environment. Biocontrol agents such as *Bacillus sp.* and its putative catalase may be useful to protect pepper from anthracnose [12]. However, the development of resistant cultivars is the best long-term strategy to control the disease, and so it is a very important goal for pepper breeders. There is still little information available about the interactions between the host and the causal pathogens of pepper anthracnose [9].

Breeding for anthracnose resistance began in the early 1990s, involving some *Capsicum* species such as, *C. annuum*, *C. frutescens,* and *C. baccatum* with potential resistance traits. It suggested that the *C. baccatum* germplasm contained higher levels of resistance to anthracnose, which may prove useful as genetic resources for anthracnose resistance [13]. Marker-assisted selection is a significant tool for the breeding of pepper. Anthracnose resistance is controlled by a major resistance locus and STS marker (CaR12.2M1-CAPS) was developed [14,15]. The introgression of the resistance gene from *C. baccatum* to *C. annuum* is difficult. For example, PBC80 was introduced into *C. annuum* through a tri-species cross by using *C. chinense* as an intermediate host [16]. New crosses have been created to combine a good source of disease resistance, such as the *C. chinense* germplasm selection PBC932, with elite Indonesian OP varieties, primarily "Jatilaba", "TitSuper" and "KR-B" ("Keriting" from Bogor). In Korea, resistant varieties, "AR legend", that crossed from *C. baccatum* to *C. annuum* with embryo rescue had been developed since 2014. There were several studies focused on the introgression of anthracnose resistance into *C. annuum* to develop new varieties [17,18].

Genetic resources with excellent disease resistance are an important prerequisite for the development of elite varieties [19]. Various studies have been reported for evaluation of *C. acutaum* resistance in pepper genetic resources [13,20–22]. Similarly different methods such as, anthracnose inoculation method, wounding and non-wounding inoculation method have been reported [23]. Non-wounding inoculation could evaluate resistance to anthracnose for cuticular wax defense of the fruit. Anthracnose development as negatively related with fruit developmental stage. As well-developed fruits had more cuticular wax than less developed fruits, the cuticular wax layers of pepper fruits may play a significant role in fruit infection by *C. gloeosporioides* isolate KG13 [24].

Phenotypic and genotypic characterization of a resistance gene (AVPP0207) located on chromosome P5 of *C. annuum* (progressive line derived from PBC932) was reported against two anthracnose isolates of *C. acutatum* and *C. truncatum* [25]. The fine mapping of a major anthracnose resistance QTL AnRGO5 in *C. chinense* 'PBC932' was also reported [26]. However, the resistance genes in *C. chinense* and *C. baccatum* were differentially expressed at different fruit maturity stages. Alternatively, some recent research reported that the inheritance of anthracnose resistance is controlled by recessive genes [27]. The finding of the study revealed that, in mature green fruit, the resistance gene is the recessive gene co1, while in ripe fruit and seedlings, the recessive genes co2 and co3, respectively, are responsible for anthracnose resistance. Mahasuk et al. found that the resistance at the ripe red fruit and mature green stages is controlled by a single dominant and single recessive gene, respectively, between an intraspecific cross derived from *C. baccatum* PBC1422 and PBC80 [28].

Sources of anthracnose resistance in *C. chinense* L. and *C. baccatum* Jacq. have been reported in Asia and used as parents in breeding programs [29]. In Korea, some studies

searching for anthracnose resistance sources have been performed [13,21,30]. However, screening for inheritance of anthracnose resistance in the wild as well as the domesticated *Capsicum* species against *Colletotrichum* are still lacking, particularly for *C. scovillei* (formerly known as *C. acutatum*). The aim of this study was to find anthracnose resistant genetic resources and make these materials available for breeding purposes.

## 2. Materials and Methods

### 2.1. Plant Materials

A total of 3738 accessions used in this study originated from 112 countries (Table 1) which includes 12 *Capsicum* species such as, *C. annuum, C. baccatum, C. glabriusculum, C. chacoense, C. chinense, C. eximium, C. frutescens, C. galapagoense, C. pendulum, C. praetermissum, C. pubescens,* and *C. tovarii.* The geographic origin and passport data of the germplasm accessions were obtained from the National Agrobiodiversity Center (NAC, Jeonju, Korea). For each accession, 8 to 10 plants were planted in a greenhouse at NAC, Jeonju, Korea, and their genetic uniformity and fruit characteristics was evaluated. The plants were irrigated with standard cultivation method made by the Rural Development Administration (RDA, Jeonju, Korea). Fully grown green fruit (approximately 30 days after pollination) were used for the post-harvest inoculation test. In the experiments, *C. annuum* 'Manitta' (Nongwoobio Co.) was used as a susceptible control, and 'AR- Dolgyeoktan' (Pepper and breeding Co.), PBC81 and PI594137, were used as resistant control [21].

**Table 1.** Origin distribution of *Capsicum* genetic resources for evaluation of resistance against anthracnose disease in this study.

| Continent | No. of Accession | Country * |
|---|---|---|
| South America | 885 | ARG, BOL, BRA, CHL, COL, ECU, GUY, PER, PRI, PRY, SUR, URY, VEN |
| North America | 585 | BHS, BLZ, CAN, CRI, CUB, GRD, GTM, HND, JAM, MEX, NIC, PAN, SLV, USA, VIR |
| Asia | 1185 | AFG, ARM, AZE, BGD, BTN, CHN, GEO, IDN, IND, IRN, IRQ, ISR, JPN, KAZ, KGZ, KHM, KOR, LAO, LKA, MDV, MMR, MNG, MYS, NPL, PAK, PHL, PRK, SYR, THA, TJK, TKM, TUR, TWN, UZB, VNM, YEM |
| Africa | 61 | MAR, BFA, BWA, DZA, EGY, ETH, GAB, GIN, KEN, LBY, MWI, NGA, SDN, SEN, TUN, TZA, UGA, ZAR, ZMB |
| Oceania | 9 | AUS, FJI, PNG |
| Europe | 724 | SUN, AUT, BEL, BGR, BLR, CHE, CSK, CZE, DEU, DNK, ESP, FRA, GBR, GRC, HUN, ITA, MDA, NLD, PRT, ROM, RUS, SRB, SVK, UKR, YUG |
| Unknown | 289 | |
| Total | 3738 | |

* Abbreviation: see Appendix A.

### 2.2. Inoculum Preparation

The fungal (*C. acutatum*) isolate 'KSCa-1' was obtained from Lee et al. [14] and the culture inoculum preparation was followed the procedures of Kim et al. [21]. The isolates were grown on potato dextrose agar (PDA) plates (Sigma Chemical Co., St. Louis, MO, USA) at 28 °C under 16 h fluorescent light/8 h dark in a temperature controlled incubation chamber. Seven-day-old PDA plates were flooded with distilled water, and fungal cultures were gently scraped from the plates. Inoculum density was adjusted to $1.0 \times 10^5$ conidia/mL with a hemacytometer.

### 2.3. Inoculation Method

Two different artificial inoculation such as, post-harvest wound and non-wound inoculations methods were used. For non-wound inoculation, fully grown green pepper

fruits 10 per accessions were kept in a re-sealable plastic bag (25 × 30 cm) with paper towel. The fruit surface was sprayed with inoculum containing anthracnose spores adjusted to a concentration of $1.0 \times 10^5$ conidia/mL. Inoculated peppers were sealed immediately and kept in thermostat at 28 °C in order to maintain the humidity for disease induction. After two days of incubation, the re-sealable plastic bags opened at room temperature for 2 h to prevent the corruption due to excessive humidity and incubated again for 14 days under the same conditions.

In wound inoculation method, microinjection by using a gas-tight micro-syringe and needle with adjustable wounding depth was used. The detached fruits 10 per accessions were washed with distilled water and on the epidermis, one to five sites according to fruit sizes were injected with 10 µL of conidial suspension containing $1.0 \times 10^5$ conidia/ml. The inoculated fruits were placed in acrylic boxes moistened with four layers of wet paper towel. The boxes were tightly sealed to maintain more than 95% relative humidity and were incubated at 28 °C with 16 h light period of up to 48 h. Finally, the boxes were uncovered and incubated again for 10 days under the same conditions.

### 2.4. Disease Evaluation

The percentage of infected sites was calculated to evaluate the disease severity of non-wounding with an average of 14 days after inoculation (Figure 1a). Based on the score, a disease rating scale (DRS) was established from 0 to 4: 0 = no symptoms; 1 = symptoms with <10% disease incidence; 2 = symptoms with 11–20%; 3 = symptoms with 21 –40%; 4 = symptoms with 41–100 % disease incidence. The phenotypes with a mean disease rating scale of <1 were evaluated as resistant (R), 1–2 as moderately resistant (MR); 2–3 as susceptible (S); and 4 as highly susceptible (HS).

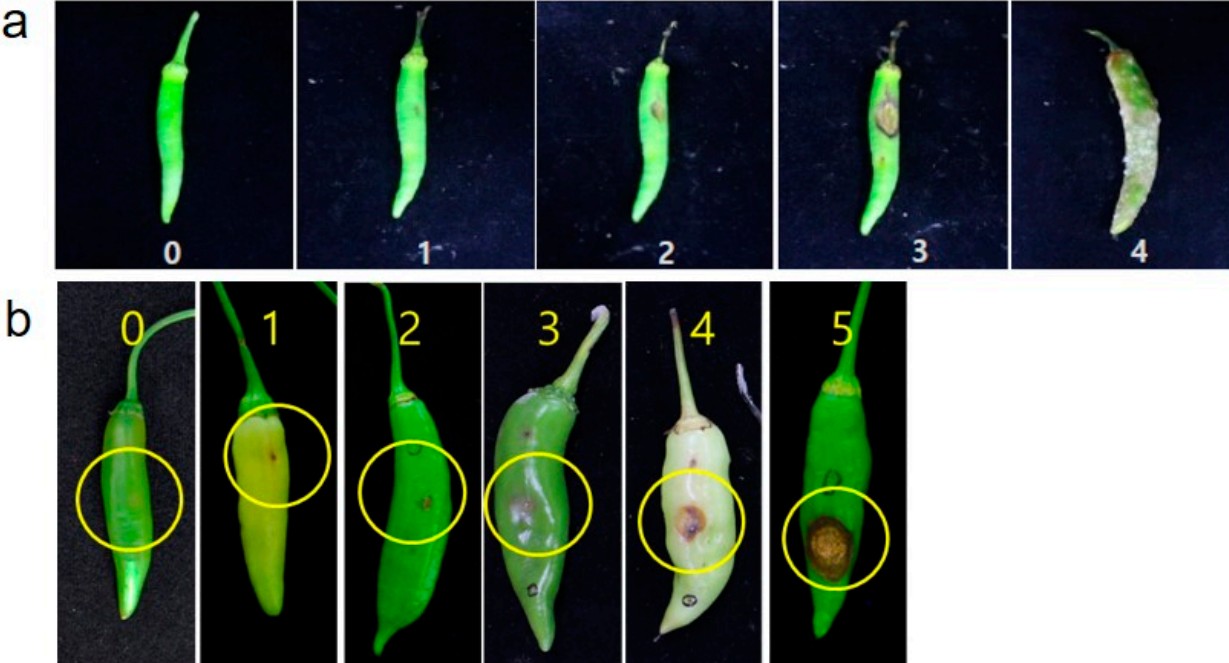

**Figure 1.** Disease severity of *C. acutatum* according to symptoms of each inoculation method. (**a**) Disease rating scale (0–4) of non-wounding inoculation (spray) and (**b**) Disease rating scale (0–5) of wound inoculation (microinjection).

Similarly, the disease severity of wound inoculation was scored finally at 2 weeks of post inoculation (Figure 1b) based on the disease rating scale 0–5 where 0 = no visible symptoms observed; 1 = symptoms of a size less than 2 mm; 2 = symptoms of a size less than 4 mm; 3 = symptoms of a size less than 6 mm; 4 = symptoms of a size less than 10 mm; 5 = more than 10 mm often encircling with acervuli as reported by Kim et al. [21].

Phenotypes with a mean disease rating scale of <1 were evaluated as resistant (R), 1–2 as moderately resistant (MR); 2–4 as susceptible (S); and >4 as highly susceptible (HS).

## 2.5. DNA Extraction and HRM Maker Analysis

Genomic DNA was extracted using the CTAB method [13]. The CA12g19240-HRM marker [31] and the CaR12.2M1-CAPS marker [15] were used to analyze the pepper genetic resources. The CA12g19240-HRM marker analysis was performed in a total reaction volume of 20 μL, containing 10 ng of genomic DNA, 2.0 μL of 10 × Taq buffer (Bioneer Co., Daejeon, Korea), 1.0 μL of 2.5 mM dNTP mixture, 0.1 μL of Taq DNA polymerase (Bioneer Co.), 1.0 μL of SYTO® 9 green fluorescent nucleic acid stain (Life Technologies™, Carlsbad, CA, USA), 0.5 μL each of 10 pmol μL$^{-1}$ of a pair of primers [32], and autoclaved distilled water for the remainder of the volume. HRM was analyzed using the LightCycler® 96 Real-Time PCR System (Roche, Basel, Switzerland) as follows; initial denaturation at 95 °C for 5 min; denaturation at 95 °C for 10 s, and annealing and elongation at 60 °C for 20 s, repeated 40 times; and final denaturation at 95 °C for 10 s; and HRM was analyzed at each temperature during a rise of 0.3% from 60 to 90 °C. The HRM graphs were drawn by LightCycler® 96 software ver. 1.1 (Roche). The CaR12.2M1-CAPS marker analysis was performed with PCR program as follows: an initial denaturation at 95 °C for 5 min; 40 cycles of amplification, each consisting of 95 °C for 45 s, 66 °C for 45 s, and 72 °C for 1 min; and a final extension at 72 °C for 5 min. The PCR production CAPS analysis was digested with the restriction enzyme *Bgl*II, and then separated on 1.2% agarosegels.

## 2.6. Fruit Characterization and Statistical Analyses

The fruit qualitative characters observed were fruit color at immature and mature stage, fruit shape, fruit color at mature stage, and fruit surface, while fruit quantitative traits including fruit length and width, fruit wall thickness, sugar contents and fruit weight were measured 3 per accession. These characters were measured at various growth stages using the standard descriptors for *Capsicum* developed by RDA [33]. Qualitative and quantitative data were analyzed using Microsoft Excel (version 16.0, Microsoft, Redmond, WA, USA). Descriptive statistics were performed using R Program (Version 4.0.2).

## 3. Results

### 3.1. Pepper Germplasm against C. acutatum with Non-Wound Inoculation

Evaluation of fungal disease resistance against the *C. acutatum* was conducted with a total of 3738 pepper germplasm. Based on the non-wounding inoculation, 261 pepper germplasm accessions showed disease resistance against *C. acutatum* infection (Table 2). Among the tested germplasm resources, 51 accessions were belonging to *C. annuum* and *C. annuum* var. *annuum*, 32 accessions were *C. baccatum*, *C. baccatum* var. *baccatum*, and *C. baccatum* var. *pendulum*, 84 accessions were *C. chinense*, and 86 accessions were *C. frutescens* of domesticated species. In *C. chacoense,* three and two accessions were showed as resistant and MR, respectively, and the other accession were susceptible to the infection. Similarly, two *C. pubescens* accessions were exhibited as MR and the remaining accessions turned out to be susceptible to the disease. The tested *C. eximium* and *C. galapagoense* accessions were showed susceptible, whereas the *C. tovarii* accession was showed resistant to the disease infection. Based on non-wounding inoculation, there were no significant difference between domesticated and wild species in disease resistance.

The development of anthracnose symptoms on susceptible and resistant Capsicum accessions by non-wounding spray inoculation with a concentration of $1.0 \times 10^5$ conidia/mL is shown in Figure 2. The typical anthracnose disease symptoms were developed on accessions of Capsicum spp., after 14 days of inoculation. The initial sunken symptoms of anthracnose were formed seven days after inoculation on susceptible accessions and increased the lesions size on fruits.

**Table 2.** Distribution of disease rating scale among the tested pepper germplasm resources against *C. acutatum* by non-wounding inoculation.

| Species | Disease Rating Scale | | | | Total |
|---|---|---|---|---|---|
| | 0–1 (R) | 1–2 (MR) | 2–3 (S) | 3–4 (HS) | |
| *C. annuum* | 2 | 3 | 8 | 684 | 697 |
| *C. annuum* var *annuum* | 49 | 27 | 26 | 1213 | 1315 |
| *C. annuum* var *glabriusculum* | 1 | - | - | 3 | 4 |
| *C. baccatum* | 18 | 12 | 6 | 55 | 91 |
| *C. baccatum* var *baccatum* | 9 | 8 | 15 | 136 | 168 |
| *C. baccatum* var *pendulum* | 5 | 15 | 6 | 97 | 123 |
| *C. baccatum* var *praetermissum* | - | - | - | 4 | 4 |
| *C. chacoense* | 3 | 2 | - | 12 | 17 |
| *C. chinense* | 84 | 43 | 40 | 512 | 679 |
| *C. eximium* | - | - | - | 1 | 1 |
| *C. frutescens* | 86 | 35 | 36 | 308 | 465 |
| *C. galapagoense* | - | - | - | 2 | 2 |
| *C. pubescens* | - | 2 | 2 | 25 | 29 |
| *C. tovarii* | 1 | - | - | - | 1 |
| *C. sp.* | 3 | - | 1 | 138 | 142 |
| Total | 261 | 147 | 140 | 3190 | 3738 |

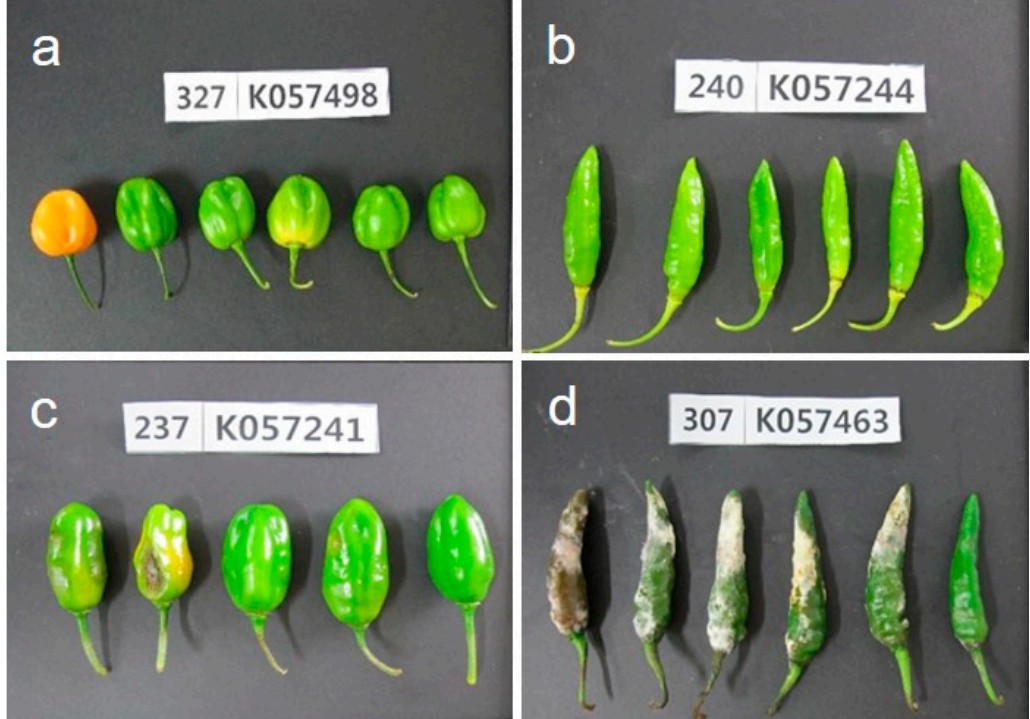

**Figure 2.** Disease symptoms of fruits after 14 days of non-wounding inoculation with *C. acutatum* isolate 'KSCa-1'. (**a**) resistant reaction of *C. chinense*; (**b**) resistant reaction of *C. baccatum*; (**c**) moderate resistant reaction of *C. chinense*; and (**d**) susceptible reaction of *C. annuum*.

### 3.2. Pepper Germplasm against C. acutatum with Wound Inoculation

To evaluates the fungal disease resistance against the *C. acutatum*, 215 pepper genetic resources were treated with wound inoculation method (Table 3). Based on non-wounding inoculation, 261 accessions were appeared resistant to the fungal disease in which 215 accessions were selected for wounding inoculation. Among the 215 tested germplasm, seven accessions of *C. baccatum*, *C. baccatum* var. *baccatum*, and *C. baccatum* var. *pendulum*, four accessions of *C. chinense* and a single accession of *C. frutescens* were

appeared with less than 25% disease incidence. Based on wound inoculation, all the selected *C. annuum* and *C. annuum* var. *annuum* accessions were appeared with more than 50% disease incidence.

**Table 3.** Distribution of disease incidence among the selected pepper germplasm resources against *C. acutatum* by wound inoculation.

| Species | Disease Incidence (%) | | | | Total * |
|---|---|---|---|---|---|
| | 0–25 | 25–50 | 50–75 | 75–100 | |
| *C. annuum* | - | - | 1 | 2 | 3 [ab] |
| *C. annuum* var. *annuum* | - | - | - | 5 | 5 [ab] |
| *C. baccatum* | 3 | 3 | 9 | 9 | 24 [b] |
| *C. baccatum* var. *baccatum* | 2 | 5 | 12 | 10 | 29 [ab] |
| *C. baccatum* var. *pendulum* | 2 | 6 | 5 | 9 | 22 [b] |
| *C. chacoense* | - | - | - | 5 | 5 [a] |
| *C. chinense* | 4 | 5 | 20 | 79 | 108 [ab] |
| *C. frutescens* | 1 | 1 | 7 | 8 | 17 [ab] |
| *C. pubescens* | - | - | 1 | - | 1 [ab] |
| *C.* sp. | - | - | 1 | - | 1 [-] |
| Total | 12 | 20 | 56 | 127 | 215 |

* Means the same letter are not significantly different in Duncan's multiple range test ($p \leq 0.05$).

After 3 days of microinjection based wound inoculation, the pin picking area began to sunken with necrotic tissues formed water-soaked lesions, which makes concentric rings of acervuli. In the wound inoculation, the typical anthracnose disease symptoms were developed on the accession of *Capsicum* spp. at 5~7 days after inoculation (Figure 3). The results of Duncan's test on disease incidence rates with 214 *Capsicum* resources excluding a *Capsicum* spp. revealed that the accessions of *C. chacoense* formed a single group (Table 3). Similarly, the accessions of *C. bacatum* and *C. baccatum* var. *pendulum* were divided into individual group, whereas the *C. baccatum* var. *baccatum*, *C. chinense*, *C. frutescens*, *C. pubescens*, *C. annuum*, and *C. annuum* var. *annuum* accessions were grouped together.

### 3.3. Maker Validation

Based on the inoculation experiment results, 261 resistant accessions were selected for marker validation. The base substitution for the anthracnose resistant (R) to susceptible (S) marker type was A → G, which was analyzed with the selected accessions. The SNP locus was converted to CAPS marker for detecting potential diversity in the *Capsicum* accessions (Table 3). The amplified SNP loci was analyzed with restriction enzyme *Bgl*II (NEB; Ipswich, England), in which SNP variations were observed with specific restriction fragment patterns (Figure 4). The results confirmed that CaR12.2M1-CAPS marker had four different restriction profiles as resistant (R), susceptible (S), heterozygous (H) and Unidentified type (UT). In order to confirm the diversity of the SNP loci among the accession, 15 accessions were selected with varying disease index (0–5), which confirmed the four different restriction profiles as resistant (R), susceptible (S), heterozygous (H) and Unidentified type (UT) or not detection (Table 4). In the CA12g19240 HRM analysis, the SNP locus showed a unique melting curve, and each accession was stratified into three groups (Figure 4) as resistant (R), susceptible (S), and heterozygous (H).

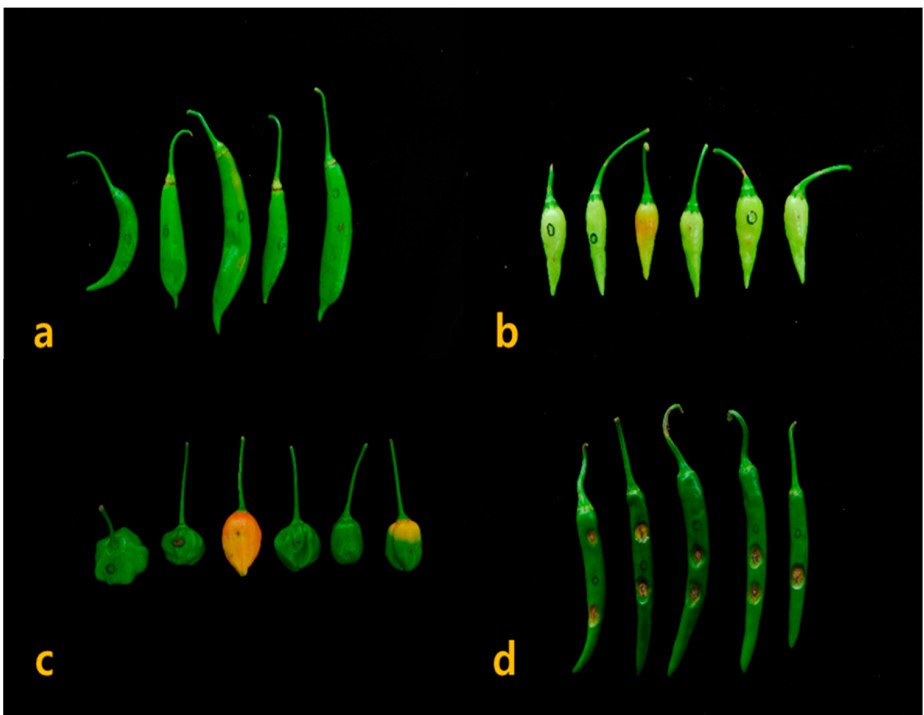

**Figure 3.** Anthracnose symptoms on resistant and susceptible *Capsicum* accessions. Resistant response of (**a**) *C. baccatum* and (**b**) *C. chinense*; Moderate resistant response of (**c**) *C. chinense* and Susceptible response of (**d**) *C. annuum*.

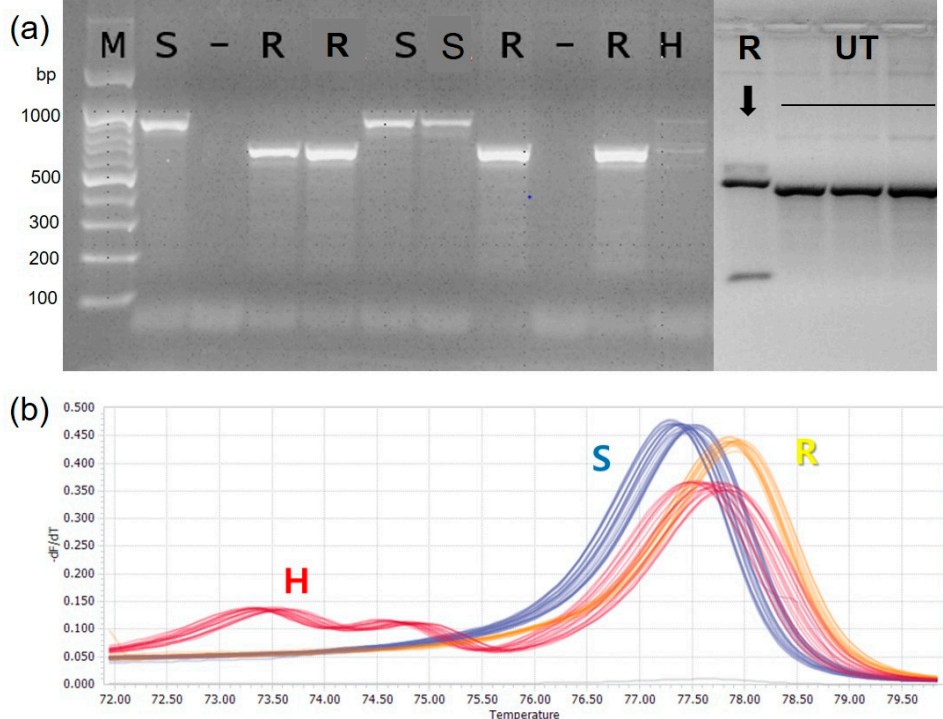

**Figure 4.** Examples of CaR12.2M1-CAPS Marker (**a**) and CA12g19240 HRM marker analysis (**b**). The CaR12.2M1-CAPS Marker analysis showed four types of genotype: resistant (R), susceptible (S), heterozygous (H) and Unidentified type (UT) along with 100 bp DNA size marker (M). The normalized melting peaks generated by CA12g19240 HRM marker analysis showed three types of genotypes: resistant (R), susceptible (S) and heterozygous (H).

**Table 4.** Incidence and Reaction to CA12g19240 and CaR12.2M1-CAPS marker of Pepper genetic resources selected with non-wound and wound inoculation.

| Acc. No. | Species | Incidence (%) | Lesion (mm) | Disease Rating Scale | CA12g19240 | CaR12.2M1-CAPS |
|---|---|---|---|---|---|---|
| 158502 | *C. chinense* | 14.3 | 1.4 ± 0.5 | 1 | S/- | S/Ud |
| 158769 | *C. baccatum* var. *pendulum* | 22.2 | 2.0 ± 0.7 | 1 | R | R |
| 218958 | *C. baccatum* var. *baccatum* | 16.7 | 1.6 ± 0.6 | 1 | R | R |
| 229147 | *C. baccatum* var. *baccatum* | 20 | 2.0 ± 1.2 | 1 | R | R |
| 229200 | *C. chinense* | 11.1 | 0.9 ± 0.3 | 1 | S | S |
| 240869 | *C. baccatum* | 15.8 | 1.5 ± 0.7 | 1 | R | R |
| 258953 | *C. baccatum* | 22.8 | 3.9 ± 1.3 | 2 | R | R |
| 270479 | *C. chinense* | 10 | 0.3 ± 0.2 | 1 | S | S |
| 276470 | *C. frutescens* | 25 | 5.3 ± 1.5 | 3 | S/- | S/Ud |
| 305437 | *C. chinense* | 20 | 0.5 ± 0.4 | 1 | S | S |
| 305455 | *C. chinense* | 10 | 0.3 ± 0.2 | 1 | S | S |
| 305478 | *C. baccatum* | 10 | 0.5 ± 0.3 | 1 | R | R |
| Manitta | *C. annuum* | 100 | 13.7 ± 2.1 | 5 | S | S |
| PBC81 | *C. baccatum* var. *pendulum* | 40 | 6.8 ± 1.2 | 4 | R | R |
| AR-Dolgyeoktan | *C. sp.* | 42 | 6.9 ± 1.6 | 4 | H | H |

Resistant (R), susceptible (S), Unidentified type (Ud), heterozygous (H) and '-' is not detection.

### 3.4. Fruit Characteristics

Five different fruit characteristics such as fruit weight(g), length(cm), width(mm), wall thickness(mm) and sugar content (ºBrix) were investigated to use the selected pepper genetic resources for breeding materials (Table 5). Three to five fruits were surveyed from each resources. Fruit color and shape of the selected pepper genetic resources showed in Figure 5.

**Table 5.** Fruit characteristics of selected anthracnose resistant Pepper genetic resources.

| Acc. No. | Origin | Species | Fruit Weight(g) | Fruit Length(cm) | Fruit Width(mm) | Fruit wall Thickness(mm) | Sugar Content (ºBrix) |
|---|---|---|---|---|---|---|---|
| 158502 | PER | *C. chinense* | 21.3 ± 0.8 | 10.2 ± 0.3 | 26.4 ± 0.8 | 3.0 ± 0.5 | 9.1 ± 0.2 |
| 158769 | CHL | *C. baccatum* var. *pendulum* | 22 ± 1.6 | 7.9 ± 0.8 | 26.5 ± 0.7 | 1.5 ± 0.1 | 12.9 ± 0.3 |
| 218958 | VEN | *C. baccatum* var. *baccatum* | 15.9 ± 1.8 | 9.9 ± 0.4 | 27.1 ± 0.6 | 1.7 ± 0.1 | 15.8 ± 0.6 |
| 229147 | HUN | *C. baccatum* var. *baccatum* | 6.4 ± 0.7 | 2.4 ± 0.1 | 21.2 ± 0.8 | 2.4 ± 0.1 | 8.5 ± 0.6 |
| 229200 | HUN | *C. chinense* | 7.2 ± 0.7 | 6.3 ± 0.3 | 22.2 ± 0.8 | 1.9 ± 0.1 | 7.6 ± 0.1 |
| 240869 | BRA | *C. baccatum* | 7.7 ± 0.5 | 5.8 ± 0.2 | 12.0 ± 2.2 | 2.0 ± 0.5 | 11.8 ± 0.9 |
| 258953 | UNK | *C. baccatum* | 1.4 ± 0.1 | 2.6 ± 0.1 | 8.4 ± 0.5 | 1.0 ± 0.1 | 10.3 ± 0.7 |
| 270479 | BRA | *C. chinense* | 1.3 ± 0.1 | 1.1 ± 0.1 | 14.1 ± 0.1 | 1.4 ± 0.2 | 8.4 ± 0.9 |
| 276470 | CRI | *C. frutescens* | 6.7 ± 0.1 | 3.2 ± 0.1 | 21.8 ± 0.6 | 1.6 ± 0.3 | 9.3 ± 0.4 |
| 305437 | COL | *C. chinense* | 8.8 ± 0.6 | 4.8 ± 0.4 | 23.2 ± 1.3 | 2.4 ± 0.2 | 7.8 ± 0.4 |
| 305455 | COL | *C. chinense* | 4.5 ± 1.0 | 4.5 ± 0.4 | 22.2 ± 2.2 | 1.5 ± 0.1 | 10.2 ± 0.5 |
| 305478 | PER | *C. baccatum* | 5.4 ± 0.2 | 4.5 ± 0.1 | 18.5 ± 1.3 | 2.7 ± 0.5 | 10.8 ± 0.3 |

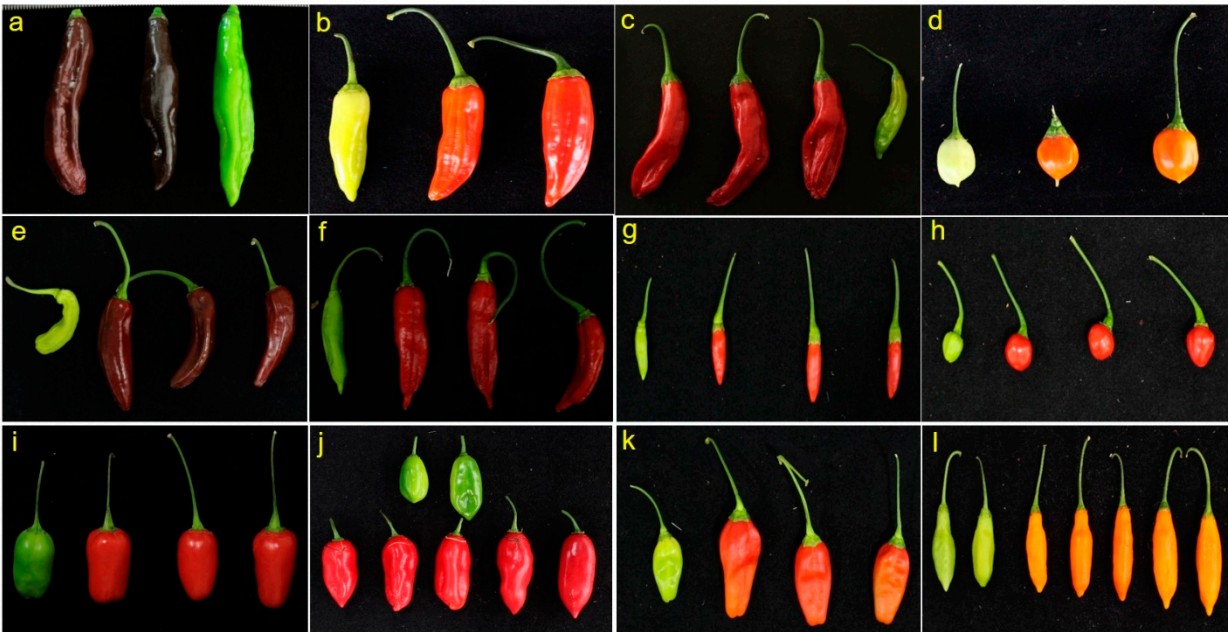

**Figure 5.** Fruit characteristics of anthracnose resistant pepper genetic resources selected for the inoculation study. (**a**) *C. chinense*; (**b**) *C. baccatum* var. *pendulum*; (**c**) *C. baccatum* var. *baccatum*; (**d**) *C. baccatum* var. *baccatum*; (**e**) *C. chinense*; (**f**) *C. baccatum*; (**g**) *C. baccatum*; (**h**) *C. chinense*; (**i**) *C. frutescens*; (**j**) *C. chinense*; (**k**) *C. chinense*; (**l**) *C. baccatum*.

## 4. Discussion

In the recent years, pepper anthracnose disease is becoming a major threat to Korean pepper production [34]. Evaluation of germplasm resources to find different resistance traits is an effective control method for Anthracnose (*Colletotrichum* spp.) disease resistance [14], which has high level of genetic diversity with different species and strains in the Korean regions [34]. Similarly, selection of plants carrying resistance genes are prerequisite for breeding studies. In pepper, molecular markers linked with anthracnose resistance genes have been identified and used in breeding programs [19]. Mainly two pepper cultivars sources are known to play a role in Anthracnose (*Colletotrichum* spp.) disease resistance [30]. The Korean genebank preserves about 6700 peppers collected from countries around the world. This study was conducted to determine the degree of resistance through non-wound and wound inoculation of *C. acutatum* on pepper fruit, and the selected resources were validated with molecular marker.

Based on the inoculation experiment, 261 resistant pepper genetic resources were selected successfully. In non-wound inoculation, *C. baccatum* and *C. chinense*, *C. chacoense* and *C. frutescens* were showed highly resistant to disease infection as reported previously [13,21]. The results of wound inoculation showed that the resistance resources were significantly distributed in the *C. baccatum* species when compared with other resources (Table 3). The *C. baccatum* species is known to exhibit stable resistance to various *Colletotrichum* spp. including *C. capsici*, *C. gloeosporioides*, *C. acutatum* species [28,35].

However, in wound inoculation, all the 215 tested accessions were developed anthracnose symptoms. Hence based on the inoculation experiments, wound inoculation method showed efficient for resistance evaluation, in which 12 resources with disease incidence rate of 0–25% were selected. The results showed there were some accessions with high incidence rates in wound inoculation, where they appeared strong resistance to non-wound inoculation. This is because the pepper anthracnose unable invades cell wall cuticle, resistance by initial defense such as cuticle layer and cell wall of plants [24,36]. It involves dynamic changes in the epidermis of the plant during pathogen infection, the crosstalk of various hormonal signaling pathways and cuticles for plant cell wall and

plant disease resistance, and the major biochemical, molecular and cellular mechanisms responsible for the role of the cuticle during plant-pathogen interactions [37].

Plants recognize the attachment of pathogens and respond very quickly by inducing the innate immunity with microbe/pathogen associated molecular pattern [37]. DAMP, a product of pathogen-infected plant degradation, such as cutin monomers and cell wall oligosaccharides, also serves as a signal to activate plant defense against pathogens [38,39]. For instance, tomato fruit cuticle was changed in response to infection with the fungal pathogen *C. gloeosporioides*, and fruit cuticle biosynthesis was upregulated during appressorium formation even before penetration [40]. Similarly, during infection of citrus by *C. acutatum*, epidermal cells responded to pathogens by increasing lipid synthesis and deposition of cuticles and cell wall-related compounds, which eventually altered the cuticle structure [41]. The *C. gloeosporioides* induced the methyl jasmonate esterase was reported [42]. Jasmonates (JAs) has been demonstrated to be involved in plant resistance to pathogens by activating pathogenic related (PR) proteins such as PR-1, PR-3, and PR-8 [43]. Some PR genes were activated, including genes encoding pathogenesis-related protein 1 and a second pathogenicity-related protein [42].

Based on these findings, pepper resources that appear to be resistant in non-wound inoculation but to be susceptible in wound inoculation could have developed cuticles or defense signaling by JAs. However, wound inoculation method skipped up the step of the cuticle's defense mechanism, it would not be able to show resistance if there were no *C. acutatum* resistance genes inside the pepper genetic resources. The 12 resources that appeared as resistant resources in wound and non-wound inoculation are likely to have two defense mechanisms. First, the cuticles are used to defend against it, and the second is the resistance genes. Anthracnose resistance is controlled by a major resistance locus [15] and resistance to *C. capsici* in 'PBC932' was found to be controlled by a single recessive gene [34]. Resistance to *C. acutatum* derived from *C. chinense* 'PBC932' was stated to be controlled by two complementary dominant genes in green fruit, but by two recessive genes in red fruit [44]. Zhao et al. (2020) narrowed down the interval of a QTL AnRGO5 conferring resistance with fine-mapping analyses [26]. Based on these findings, it can be seen that the major genes or QTL involved in *Capsicum* anthracnose are present according to a specific pathogen and *Capsicum* species.

As a result of testing with CA12g19240 marker and CaR12.2M1-CAPS marker from selected accessions (Table 4), the markers for *C. baccatum* and *C. annuum* were well matched, but *C. chinense* and *C. frutescens* were susceptible although their phenotype to *C. acutatum* were resistant. As CA12g19240 marker and CaR12.2M1-CAPS marker were developed between *C. baccatum* and *C. annuum*, two makers could not distinguish resistance to *C. acutatum* in *C. chinense* and *C. frutescens*. Recently, a marker using PBC932 was developed [26]. However, it has not yet been applied to *C. chinense* genetic resource, it is expected that *C. chinense* can be used to determine resistance using a marker derived from PBC932. The CAPS and HRM markers have been reported to detect the intra- and interspecies variation and genotypic discrimination of different species [45]. Similarly, in this study CAPS and HRM markers have successfully identified the diversity between the *Capsicum* species.

The *Capsicum* species has been studied using morphological as well as with molecular markers [46]. The genetic similarity information can complement phenotypic information in the development of breeding populations [47]. Thus, morphological characterization is an important step in the classification of germplasm. Previously, Luitel et al. reported wide variation in the pepper fruit characters of a core collection [48]. Similarly, in this study the fruit characteristic analysis revealed that the selected pepper fruits vary in size, shape, color and even in their sugar content as the pepper germplasm collected from different countries. Since each country has different pepper preferences, selecting various anthracnose resistant genetic resources will help international pepper breeding.

## 5. Conclusions

As being attempted worldwide to breed *Capsicum* for anthracnose resistant, it is important to evaluate various pepper germplasm and select excellent pepper genetic resources. In this study, a total of 261 accessions were selected which showed as resistance for non-wound inoculation. By selecting and testing them with wound inoculation experiment, 12 accessions were showed less than 25% of the disease incidence, indicating resistance. However, the CAPS and HRM markers analysis showed diversity of alleles, and hybrids in the tested accessions. Therefore, the findings of this study might provide useful information for understanding the genetic variability of tested accessions. Further, the validation of the CAPS and HRM markers which linked to anthracnose resistance revealed steady association of the marker in anthracnose-resistant accessions. These pepper genetic resources are expected to be used as materials for anthracnose-resistant breeding and genetic studies.

**Author Contributions:** Conceptualization, N.-Y.R. and B.-C.K.; methodology, O.-S.H. and B.G.; investigation, Y.-J.L. and R.S.; resources, G.-T.C.; data curation, N.-Y.R.; writing—original draft preparation, N.-Y.R.; writing—review and editing, R.S.; supervision, B.-C.K.; project administration, N.-Y.R. All authors have read and agreed to the published version of the manuscript.

**Funding:** This research received no external funding.

**Institutional Review Board Statement:** Not applicable.

**Informed Consent Statement:** Not applicable.

**Acknowledgments:** Financial support for the study was obtained from Research Program of Agricultural Science and Technology Development (Project No. PJ013251022020), National Institute of Agricultural Sciences, RDA, Republic of Korea.

**Conflicts of Interest:** The authors declare no conflict of interest.

## Appendix A

Abbreviations of country names.

| Appendix A. Abbreviations of Country Names. | |
| :---: | :---: |
| **Abbreviation** | **Nation** |
| 9 | Europe |
| 27 | South America |
| AFG | Islamic State of Afghanistan |
| ARG | Argentine Republic |
| ARM | Republic of Armenia |
| AUS | Australia |
| AUT | Republic of Austria |
| AZE | Azerbaijani Republic |
| BEL | Kindgom of Belgium |
| BFA | Burkina Faso |
| BGD | People's Republic of Bangladesh |
| BGR | Republic of BμL garia |
| BHS | Commonwealth of the Bahamas |
| BLR | Republic of Belarus |
| BLZ | Belize |
| BOL | Republic of Bolivia |
| BRA | Federative Republic of Brazil |
| BTN | Kingdom of Bhutan |
| BWA | Republic of Botswana |
| CAN | Canada |

| Appendix A. Abbreviations of Country Names. | |
| --- | --- |
| **Abbreviation** | **Nation** |
| CHE | Swiss Confederation |
| CHL | Republic of Chile |
| CHN | People's Republic of China |
| COL | Republic of Colombia |
| CRI | Republic of Costa Rica |
| CSK | Czechoslovakia |
| CUB | Republic of Cuba |
| CZE | Czech Republic |
| DEU | Federal Republic of Germany |
| DNK | Kingdom of Denmark |
| DZA | People's Democratic Republic of Algeria |
| ECU | Republic of Ecuador |
| EGY | Arab Republic of Egypt |
| ESP | Kingdom of Spain |
| ETH | Ethiopia |
| FJI | Republic of Fiji |
| FRA | French Republic |
| GAB | Gabonese Republic |
| GBR | United Kingdom of Great Britain and Northern Ireland |
| GEO | Republic of Georgia |
| GIN | The Republic of Guinea |
| GRC | The Hellenic Republic |
| GRD | Grenada |
| GTM | Republic of Guatemala |
| GUY | Cooperative Republic of Guyana |
| HND | Republic of Honduras |
| HUN | Republic of Hungary |
| IDN | Republic of Indonesia |
| IND | Republic of India |
| IRN | Islamic Republic of Iran |
| IRQ | The Republic of Iraq |
| ISR | State of Israel |
| ITA | The Italian Republic |
| JAM | Jamaica |
| JPN | Japan |
| KAZ | Republic of Kazakhstan |
| KEN | The Republic of Kenya |
| KGZ | Kyrgyz Republic |
| KHM | Cambodia |
| KOR | Republic of Korea |
| LAO | Lao People's Democratic Republic |
| LBY | Socialist People's Libyan Arab Jamahiriya |
| LKA | Democratic Socialist Republic of Sri Lanka |
| MAR | Kingdom of Morocco |
| MDA | Republic of Moldova |
| MDV | Republic of Maldives |
| MEX | United Mexican States |
| MMR | Union of Myanmar |
| MNG | Mongolia |
| MWI | Republic of Malawi |
| MYS | Malaysia |
| NGA | Federal Republic of Nigeria |
| NIC | Republic of Nicaragua |
| NLD | Kingdom of the Netherlands |

| Appendix A. Abbreviations of Country Names. | |
| --- | --- |
| **Abbreviation** | **Nation** |
| NPL | Kingdom of Nepal |
| PAK | Islamic Republic of Pakistan |
| PAN | Republic of Panama |
| PER | Republic of Peru |
| PHL | Republic of the Philippines |
| PNG | Papua New Guinea |
| PRI | Puerto Rico |
| PRK | Democratic People's Republic of Korea |
| PRT | Portuguese Republic |
| PRY | Republic of Paraguay |
| ROM | Romania |
| RUS | Russian Federation |
| SDN | Republic of the Sudan |
| SEN | Republic of Senegal |
| SLV | Republic of El Salvador |
| SRB | Republic of Serbia |
| SUN | Union of Soviet Socialist Republics |
| SUR | Republic of Suriname |
| SVK | Slovak Republic |
| SYR | Syrian Arab Republic |
| THA | Kingdom of Thailand |
| TJK | Republic of Tajikistan |
| TKM | Turkmenistan |
| TUN | Republic of Tunisia |
| TUR | Republic of Turkey |
| TWN | Taiwan Province of China |
| TZA | United Republic of Tanzania |
| UGA | Republic of Uganda |
| UKR | Ukraine |
| UNK | Unknown |
| URY | Eastern Republic of Uruguay |
| USA | United States of America |
| UZB | Republic of Uzbekistan |
| VEN | Republic of Venezuela |
| VIR | Virgin Islands of the United States |
| VNM | Socialist Republic of Viet Nam |
| YEM | Republic of Yemen |
| YUG | Federal Republic of Yugoslavia |
| ZAR | Republic of Zaire |
| ZMB | Republic of Zambia |

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
