# Peer review of "Evaluation of Anthracnose Resistance in Pepper (Capsicum spp.) Genetic Resources"

_horticulturae, doi:10.3390/horticulturae7110460_

Round 1
Reviewer 1 Report
Major revisions requested. See details in attached file.

Author Response
As per the reviewer's suggestion, we have revised the manuscript. please find the attached word file

Reviewer 2 Report
Please, see comments attached in the paper.

Author Response
As per the reviewer's suggestion, we have revised the manuscript. please find the attached pdf file

Reviewer 3 Report
The authors present in paper research titled “Evaluation of Anthracnose Resistance in Pepper (Capsicum spp.) genetic resources”.
This work has the main aim of this study is to find resistant lines of Capsicum spp. from a germplasms bank. The authors evaluated with 3,738 pepper genetic resources, collected from different countries and conserved at Korean GenBank. They studied the resistance to C. acutatum under laboratory conditions by spray inoculation (non-wounding) method and using microinjection inoculation (wounding) method.
Based in my professional knowledge in the study of Colletotrichum and Anthracnose in different crops. I would like to congratulate the authors for the work presented. This type of study takes a lot of time and dedication. And finally, being able to achieve the objective of selecting a large number of resistant lines (12 in wound infection studies), from a large number of initial candidates, seems very interesting to me.
The article seems to me to be correctly written, and the experiments and results are clearly presented.
It is true that the results may not be entirely as expected, since the 12 resistant lines did not fully correlate with the genetic markers of resistance CA12g19240 and CaR12.2M1-CAPS in all the pepper species analyzed. But this does not detract from the study, quite the opposite.
This work should be continued, in order to find suitable mocular markers for the species C. chinense and C. frutescens.
I think the article should be accepted for publication, just with minor typographical errors.
Throughout the introduction Capsicum is named on many occasions without using italics (lines 36,39, 42, 43, 101).
Line 128, 135, 143.- the spore concentration must be corrected using the superscript for number 5.
Line 147, 10 must be separated from the word days.
Line 197, 362,363, 378 - Capsicum in italics.
Author Response

(The authors gave the same response as above.)

Reviewer 4 Report
Latin names should be in italics.
English in this manuscript should be improved. There are some mistakes in English expression.
The names of genes should be in italics.
The concentration in Line 135 should be corrected.
The concentration in Line 143 should be corrected.
In this manuscript, the ‘disease index’ should be replaced by ‘disease severity’. The two terms are different.
Two inoculation methods, spray inoculation (non-wounding) method and microinjection inoculation (wounding) method, were used in this manuscript. Which inoculation method could be more useful in the resistance evaluation?
Figure 1 is not clear enough.
In the References section, there are a lot of problems, and each reference should be checked according to the requirements of the Journal.
Although the reference of the sequences of the primers used for detected the markers was provided, the sequences should be provided in details in this manuscript.
The electrophoretic band sizes of the marker should be labeled in Figure 4.
Author Response

(The authors gave the same response as above.)

Round 2
Reviewer 1 Report
- The reworded abstract (lines 22 to 32) needs further improvement.
- Lines 105 to 109 should be removed and replaced with “The aim of this study was to find anthracnose resistant genetic resources and make these materials available for breeding purposes”
- ‘Results’-Genotyping should be carried out on the original 261 accessions identified to be resistant by spray inoculation (non-wounding) method and not only the 12 accessions confirmed to be resistant by the microinjection inoculation (wounding) method.
- The authors should minimize their molecular conclusions since the markers showed 50% recombination
Author Response
Comments and Suggestions for Authors
1. The reworded abstract (lines 22 to 32) needs further improvement.
Response: As per reviewer instruction, the abstract section of the manuscript has been revised.
2. Lines 105 to 109 should be removed and replaced with “The aim of this study was to find anthracnose resistant genetic resources and make these materials available for breeding purposes”
Response: As per reviewer instruction, the aim of the study information of the manuscript has been revised.
3. ‘Results’-Genotyping should be carried out on the original 261 accessions identified to be resistant by spray inoculation (non-wounding) method and not only the 12 accessions confirmed to be resistant by the microinjection inoculation (wounding) method.
Response: As per reviewer instruction, the genotyping of all 261 accessions identified in the spray inoculation (non-wounding) method has been revised. However, the relevant figure was not included in the revised version of the manuscript to avoid the repetition of Figure 4.
4. The authors should minimize their molecular conclusions since the markers showed 50% recombination
Response: As per the reviewer's suggestion, the conclusion of marker information has been revised in the manuscript.
Reviewer 2 Report
the ms has been reviewed accordingly.
We consider valuable all changes produced.
Author Response
Thank you for your valuable suggestion on our submitted manuscript.
Reviewer 4 Report
In this manuscript, the ‘disease index’ should be replaced by ‘disease severity’. The two terms are different. The disease index could be replaced by “disease rating scale”.
The names of genes should be in italics. The names of genes and those of Proteins should be distinguished.
Latin names should be in italics, especially in the References section.
The electrophoretic band sizes of the marker should be labeled in Figure 4.
Two inoculation methods, spray inoculation (non-wounding) method and microinjection inoculation (wounding) method, were used in this manuscript. Which inoculation method could be more useful in the resistance evaluation? This should be discussed in the manuscript.
Author Response
Comments and Suggestions for Authors
In this manuscript, the ‘disease index’ should be replaced by ‘disease severity. The two terms are different. The disease index could be replaced by the “disease rating scale”.
Response: As per the reviewer's suggestion, the information of the ‘disease index’ has been corrected with the “disease rating scale” in the revised manuscript.
The names of genes should be in italics. The names of genes and those of Proteins should be distinguished.
Response: As per reviewer instruction, we have re-checked the gene and protein name information in italics.
Latin names should be in italics, especially in the References section.
Response: As per journal instruction, we have used the word template file from the journal site along with reference managing software (EndNote). We hope the production team will verify and finalize the published version of the manuscript.
The electrophoretic band sizes of the marker should be labeled in Figure 4.
Response: As per reviewer instruction, we have included the band size of the marker in Figure 4.
Two inoculation methods, the spray inoculation (non-wounding) method, and the microinjection inoculation (wounding) method were used in this manuscript. Which inoculation method could be more useful in the resistance evaluation? This should be discussed in the manuscript.
Response: As per reviewer instruction, we have included information on the efficient inoculation methods for disease evaluation in the discussion section of the revised manuscript.